

**Groundwater-surface water relations in regulated lowland catchments;**
**hydrological and hydrochemical effects of a major change in surface water**
**level management**
*Joachim Rozemeijer[1*], Janneke Klein[1], Dimmie Hendriks[1], Wiebe Borren[2], Maarten*
*Ouboter[3], Winnie Rip[3]*
*[1] Deltares, P.O. Box 85467, 3508 AL Utrecht, The Netherlands*
*[2] Natuurmonumenten, P.O. Box 9955 ,1243 ZS  's Graveland, The Netherlands*
*[3]* Water Authority Amstel, Gooi en Vecht, *P.O. Box* 94370, 1090 GJ Amsterdam, *The*
*Netherlands*
* Corresponding author, tel. +3162748708, email: Joachim.rozemeijer@deltares.nl
**Abstract**
In lowland deltas with intensive land use such as The Netherlands, surface water levels are
tightly controlled by inlet of diverted river water during dry periods and discharge via large-scale
pumping stations during wet periods. The conventional water level regime in these polder
catchments is either a fixed water level year-round or an unnatural regime with a lower winter
level and a higher summer level in order to optimize hydrological conditions for agricultural land
use. The objective of this study was to assess the hydrological and hydrochemical effects of
changing the water level management from a conventional fixed water level regime to a flexible,
more natural regime with low levels in summer and high levels in winter between predefined
minimum and maximum levels.
Ten study catchments were hydrologically isolated and equipped with controlled inlet and outlet
weirs or pumping stations. The water level management was converted into a flexible regime.
We used water and solute balance modeling for catchment-scale assessments of changes in
water and solute fluxes.



Our model results show relevant changes in the water exchange fluxes between the polder
catchment and the regional water system and between the groundwater, surface water, and field
surface storage domains within the catchment. Compared to the reference water level regime,
the flexible water level regime water balance scenario showed increased surface water residence
times, reduced inlet and outlet fluxes, reduced groundwater-surface water exchange, and in
some catchments increased overland flow. The solute balance results show a general reduction
of chloride concentrations and a general increase in N-tot concentrations. The total phosphorus
(P-tot) and sulfate (SO4) concentration responses varied and depended on catchment-specific
characteristics.
For our study catchments, our analyses provided a quantification of the water flux changes after
converting towards flexible water level management. Regarding the water quality effects, this
study identified the risks of increased overland flow in former agricultural fields with nutrient
enriched top soils and of increased seepage of deep groundwater which can deliver extra
nutrients to surface water. At a global scale, catchments in low-lying and subsiding deltas are
increasingly being managed in a similar way as the Dutch polders. Applying our water and solute
balance approach to these areas may prevent unexpected consequences of the implemented
water level regimes.
**Keywords**
Water level management, Hydrology, Water quality, Water and solute balance modelling
**1      Introduction**
Water levels in the flat, low-lying polder catchments in the western and northern parts of the
Netherlands are tightly regulated. Water authorities maintain either a year-round fixed water level
or fixed summer and winter levels. During dry periods, these water levels are maintained by
diverting river water into these catchments (e.g. Roelofs, 1991, Rozemeijer, 2012). This water
inlet prevents drought damage in agriculture and wetland nature reserves. In addition, too low
water levels would accelerate peat oxidation which causes land subsidence (e.g. Schothorst,

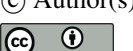



1977; Hoogland et al., 2012) and greenhouse gas emissions (e.g. Schrier-Uijl et al., 2014).
During wet conditions, the water levels are maintained to prevent water damage to agriculture,
roads, and buildings. Excess water is discharged towards the regional rivers and canal systems
via large-scale pumping stations.
The conventional water level regime in polder catchments is either a fixed water level year-round,
or a winter level that is lower than the summer level. These unnatural regimes optimize
hydrological conditions for agricultural land use. During the wet winters the drainage of
groundwater is enhanced by relatively low surface water levels. During the dry summer however,
the infiltration of surface water to replenish groundwater and soil moisture is enhanced by high
surface water levels due to the inlet of diverted river water (Hendriks et al., 2013).
Several water authorities consider changing the water level regimes to a more natural regime,
where surface water levels fluctuate freely between predefined minimum and maximum water
levels. Such a change in water level regime is often motivated from an ecological perspective.
Restoring natural water level fluctuations stimulates dispersal and germination of plant species
and increases vegetation diversity in the riparian zones (Sarneel et al., 2014). In a flexible water
level regime, temporal flooding distributes seeds over the riparian gradient. The subsequent
receding water levels create a variety of soil moisture conditions that meet the germination
requirements of many plant species (Lenssen et al., 1998; Sarneel et al., 2014). Van Leeuwen et
al. (2014) reported seed bank experiments in which the spring drawdown of water levels in a
flexible regime resulted in higher species richness and diversity compared to the conventional
regimes with fixed water levels.
An additional advantage of restoring water level fluctuations is the potential reduction of the
amount of diverted river water inlet needed from the regional water system. As long as the
minimum water level is not reached, no inlet water is needed. In addition summer storm
precipitation water does not leave the polder as long as the maximum water level is not



exceeded. The reduced need for inlet water enhances the regional water availability during
droughts and may improve water quality within the polders.
Several authors reported negative consequences of diverted river water inlet for the water quality
and ecology of receiving surface waters. For example, Delaune et al. (2005) and Miao et al.
(2011) studied eutrophication caused by the inlet of Mississippi River water into N-limited wetland
ecosystems. Roelofs (1991), Smolders et al. (2006), and Lamers et al. (2013) reported that the
inlet of alkaline ($HCO_3$ and $SO_4$-rich) river water may also enhance the release of phosphorus
from sediments in the receiving surface waters.
A final potential advantage of converting to a flexible water level regime and allowing water levels
to rise during the wet winter period is the reduction of the amount of discharge that needs to be
pumped out of the polder into the regional river system. This would reduce the consumption of
energy and the costs for maintenance of the pumping stations and reduce the pressure on the
regional water system during high discharges. Introduction of a flexible water level management
is often postponed or hampered due to the fear for negative agricultural or recreational impacts,
e.g. flooding of agricultural land in winter and spring and insufficient water depth for boating in
summer.
Although water quality is an important issue in polder catchments in Netherlands, little is known
about the combined effects of restoring natural water level fluctuations on water and solute
fluxes. Small changes in nutrient concentrations in polders can result in dominance of undesired
species like duckweed (Lemnaceae) or floating fern (Azolla), a severe reduction in biodiversity,
and hypoxia (Janse and Van Puijenbroek, 1998; Hellman and Vermaat, 2012; Vermaat et al.,
2016). The reduced impact of water inlet suggests a positive effect of flexible water level
management for chemical water quality. However, other processes may induce reverse effects in
these hydrological complex polder systems, such as the changing interaction between
groundwater and surface water. For example, higher water levels in winter may enhance the
leaching of nutrients from the enriched upper part of the soils (Rozemeijer and Broers, 2008; Van




der Grift, 2016). At the same time, lower water levels in summer may enhance seepage of
nutrient-rich and/or saline groundwater from the peaty subsurface (e.g. Van Beek et al., 2007;
Delsman et al., 2017; Yu et al., 2017). In addition, lower surface water levels in summer may
enhance peat oxidation which causes subsidence and nutrient losses. Throughout the world, an
increasing amount of catchments in low-lying and subsiding deltas will be managed in a similar
way as the Dutch polders. Therefore, the importance of understanding of the effect of different
water level regimes increases.
A combination of water and solute mass balancing has been proven effective for disentangling
the combined effect of (changes in) multiple sources and pathways of water and solutes in
similar catchments. For example, Hellman and Vermaat (2012) studied the relative impact of
agricultural nutrient inputs by constructing water and nutrient budgets for 13 polders in The
Netherlands. Kieckbusch and Schrautzer (2007) used water and nutrient balances to study the
effects of raising water levels in two catchments in Northern Germany.
The objective of this study was to assess the combined hydrological and hydrochemical effects
of changing the water level management from a fixed water level regime to a flexible, more
natural regime with minimum and maximum water levels. Ten study catchments were partly
hydrologically isolated; the surface water connections with the surroundings were closed apart
from controllable inlet and outlet weirs or pumps. The surface water level control was changed to
a flexible regime with predefined minimum and maximum water levels. The catchments were
intensively monitored in a dense multi-scale network of monitoring locations for surface water
and groundwater levels and solute concentrations in soil moisture, groundwater, and surface
water. The reference situation, the timing of the actual change in water level management, and
the practical water level management varied among the study catchments. Therefore, the
monitoring did not provide us with adequate data for comparing the situation before and after the
implementation of flexible water level management. We therefore used water and solute balance
modelling for catchment-scale assessments of changes in water and solute fluxes caused by the
major change in water level management.



# 2    Methods

## 2.1    Study area

The ten investigated polder catchments are situated in the Midwestern part of The Netherlands, between the cities of Amsterdam in the northwest and Utrecht in the southeast (see Figure 1). All polders are within the management area of water authority Amstel, Gooi en Vecht. All studied polders are below mean sea level (MSL) due to centuries of artificial drainage, land subsidence, and peat extraction. The current land surface elevations range between 5.0 m and 0.6m below MSL (Table 1). The subsurface consists of Pleistocene sands, covered with a layer of Holocene peat and clay of variable thickness (see Table1). The original thickness of this cover layer reduces from circa 8 m in the northwest (Middelpolder, Ronde Hoep) to 1 m in the southeast (Oostelijke Binnenpolder, Westbroekse Zodden) near the sandy ice-pushed ridge 'Utrechtse Heuvelrug'. However, in the Groene Jonker and in the lakes of Botshol, Loenderveen Oost, and Muyeveld, most of this Holocene peat has been mined in the 17th and 18th century to be used as fuel.

The area has a semi-humid sea climate with an average yearly precipitation of 800mm and an average yearly estimated evaporation of 550 mm, resulting in an average estimated yearly recharge of 250 mm. The area is also fed by groundwater seepage from the Utrechtse Heuvelrug. However, whether the individual polders receive seepage or infiltrate to the regional groundwater system depends on their elevation relative to neighboring polders (e.g. Oude Essink, 2001). The Ronde Hoep, for example, lies at 2.3 m below MSL, but still loses water towards polder Groot-Mijdrecht at ca. 6 m below MSL bordering at the south.

All investigated polders are reclaimed wetland nature reserves. The Middelpolder, Ronde Hoep, and parts Muyeveld are still in extensive agricultural use (grass/reed harvest, cattle grazing). Loenderveen Oost is a fallback drinking water reservoir for the city of Amsterdam. Especially Muyeveld is rich in recreational activities (boating, fishing, camping). The Groene Jonker and the Westbroekse Zodden are partly accessible for hiking and bird watching. Most other polders are



less accessible or restricted bird reserves. The studied polders differed in their percentages
covered by surface water; the surface water proportion varies between 4% and 98% (Table 1).
The polders with low surface water proportions, such as Ronde Hoep, Nieuwe Keverdijkse
polder, and Middelpolder, have been in intensive agricultural use and still have their artificial
drainage network of ditches. The catchments of Botshol, Muyeveld, and Oostelijke Binnenpolder
partly consist of drained fields and partly of open surface water lakes. Open surface water
dominates the catchments of Loenderveen Oost and Westbroekse zodden. The original ditch
network of the Groene Jonker catchment was largely removed in a wetland restoration project.
For more detailed descriptions of all studied polder catchments, we refer to Borren et al. (2012a).
The surface water levels in all catchments are human controlled through large scale pumping
stations, adjustable weirs, and/or culverts. The water level management in all catchments was
changed between 2008 and 2013 from a fixed year-round water level or fixed winter and summer
level regime into a flexible regime in which the water levels can fluctuate freely between a
minimum and a maximum level. Note that the moment of implementation of the flexible water
level regime varies quite a lot among the study catchments due to the progress of legal
processes (paperwork, stakeholder consultation) and on the planning of the infrastructural work.
Table 1 gives for each catchment the fixed water levels of the conventional reference regime and
the minimum and maximum levels of the new flexible regime. Most polders have the same or
similar average water levels in the new flexible regime, except Ronde Hoep and Nieuwe
Keverdijkse polder north. The new water levels in these polders are higher in the new situation.
No reference water level regime is given for the Groene Jonker, because the landscape was
severely reshaped before introducing the flexible water level regime.
**2.2   Monitoring**
All studied polders were intensively monitored in 2011 and 2012 in a dense multi-scale network
of measurement locations for surface water and groundwater levels and solute concentrations.
This monitoring was mainly implemented for getting a more detailed overview of water and solute





fluxes in the study catchments. We did not implement the monitoring for comparing the situation
before and after the implementation of flexible water level management. This was not possible
within the timeframe of the monitoring period (2011-2012), because the timing of the actual
change in water level management varied among the study catchments (2008-2013).
Piezometers were installed to measure phreatic and deep groundwater heads and surface water
levels. The amounts of piezometers varied from 15 in the relatively small Oostelijke Binnenpolder
to 47 in Muyeveld. Part of the piezometers were distributed along the expected regional scale
hydrological gradients and the other part were placed in field scale (5-50m) transects from the
surface water ditches or lakes, through the riparian zones into the fields. Surface water and
groundwater levels were registered at hourly time intervals using pressure sensors (van Essen
instruments, Delft, the Netherlands). More details on the water level monitoring are described by
Borren et al. (2012a).
Concentrations of nutrients and other macro-parameters were monitored through monthly
sampling of surface water, groundwater, and soil moisture (Smolders et al., 2012). In each
polder, surface water was sampled at the inlet and outlet of the catchment and at several
locations distributed over the polder. Both shallow (phreatic) and deep groundwater from below
the Holocene top layer were sampled from piezometers using peristaltic pumps. Soil moisture
was sampled from porous ceramic suction cups that were installed close to the piezometers at
depths of 25, 50, and 100 cm below the land surface. The surface water, groundwater, and soil
moisture samples were analysed for pH (pH sensor), alkalinity (titration), and elements (e.g. Na,
K, $NH_4$, $NO_3$, $PO_4$, $SO_4$, Cl, Ca, Mg, S, P, Fe, Mn, Si, Zn) using Auto-Analyzers (AA) and
Inductively Coupled Plasma Atomic Emission Spectroscopy (ICP-OES). Details on the water
quality monitoring are reported in Smolders et al. (2012).
Water quality in the regional river water that was diverted into the polders during dry periods was
monitored monthly by the regional water authority Amstel, Gooi en Vecht. Amstel, Gooi en Vecht
also monitored the inlet and outlet water fluxes of the catchments, either by registering the


pumping hours or by monitoring water levels at weirs. Solute concentrations in precipitation water
were monitored by the National Institute for Public Health and the Environment (e.g. RIVM,

3   2005).

**2.3     Water and solute balance modelling**
To assess the changes in water and solute fluxes caused by flexible water level management,
water and solute balances were set up for all polder catchments. To facilitate future application
by the water authority staff, the water and solute balances were built in MS Excel. The water
storage in the catchment was divided in a groundwater, surface water, and field storage domain
(Figure 2). All exchange fluxes between these storages and between the catchment and its
surroundings were quantified at daily time steps in the water balance model. The only time-
variable inputs were the measured precipitation and evaporation time series from nearby weather
stations operated by the Royal Netherlands Meteorological Institute (KNMI). All other fluxes were
calculated following the model formulations in Supplement A. This approach allowed for uniform
scenario analyses for all catchments, assuming that the change in water level management did
not have a large influence on precipitation and evaporation.
The water level management was represented in the water balance by fixed year-round or fixed
summer and winter water levels for the reference conventional regime (Ref) and by minimum and
maximum water levels for the flexible water level management scenarios (Flex). Exceeding these
fixed or minimum and maximum levels induced water inlet or outlet to and from the polders (see
Supplement A). In case of a pumping station outlet, a maximum outlet discharge was imposed.
The solute balance modelling focused on the inputs and outputs of Cl, P-tot, N-tot, and SO4 to
and from the surface water domain. Average measured concentrations were assigned to all input
fluxes towards the surface water (overland flow, groundwater exfiltration, seepage, precipitation,
water inlet; see Figure 2). The concentrations in overland flow were not directly measured, but
were assumed to be equal to the average measured soil moisture concentrations at 25 cm below
the surface. For groundwater exfiltration, separate concentrations were assigned to groundwater





flow from the topsoil (upper 30 cm) and groundwater flow from below the topsoil. The average
measured soil moisture concentrations at 25 cm below surface were assigned to groundwater
exfiltration from the topsoil. Groundwater exfiltration from below the topsoil was assigned the
average concentrations of samples from suction cups at 50 and 100 cm below the surface and
from the upper groundwater piezometers. The solute concentrations measured in samples from
the deep groundwater piezometers were assigned to the seepage flux. The concentrations
assigned to precipitation water came from the national monitoring network for precipitation
composition (RIVM, 2005), while concentrations in the diverted river water were provided by
water authority Amstel, Gooi en Vecht.
The water and solute balances were manually calibrated towards an optimal representation of
the dynamics in measured surface water levels, groundwater levels, Cl concentrations, and, if
available, discharge at the inlet and outlet of the polder catchments. The most sensitive
parameters were the surface water and land area percentages ($A_l$ and $A_w$ in Supplement A), the
infiltration or seepage fluxes ($F_{si}$ or $K_v$, d, and $h_d$ depending on the lower boundary definition),
and/or the exfiltrating fraction of groundwater towards surface water ($f_l$). As an example, Figure 3
shows the model performance on simulating the surface water levels, groundwater levels, and
chloride concentrations for the Middelpolder. The same information for the other polder
catchments is given in Supplement B. For more details on the water and solute balance models
we refer to Borren et al. (2012b).





**3      Results**
The results of the water balance modelling for the reference and the flexible water level
management for all catchments are summarized in Table 2, 3, and 4. Many different output
results can be obtained from the water and solute balance models (Borren et al., 2012b). We
focus here on the differences in water fluxes and solute concentrations between the reference
and the flexible water level scenarios.
Table 2 gives the calculated average residence times and the exchange fluxes over 2003-2011
between the catchments and the regional surface water system via inlet and outlet and between
the catchments and the deep groundwater system via infiltration and/or seepage. Relative to the
reference scenario, flexible water level management results in longer surface water residence
times in all catchments. The residence times increase with 8-204%, with the largest relative
increases in the catchments with the lowest surface water area percentages (Ronde Hoep,
Nieuwe Keverdijkse polder north and south). This increase in residence time is related to the
decrease in inlet and outlet volumes, while the total water volume in the polders does not change
much. In polders with relatively small water volumes (Ronde Hoep, Nieuwe Keverdijkse polder
north and south), the decrease in inlet and outlet volumes have a relatively large impact on the
residence time.
The increase in surface water residence times is directly related to the decrease in both the
amount of inlet of diverted river water and the outlet to the regional water system (Table 2), while
the total water volume in the polders does not change much. The inlet and outlet volumes are
lower in all catchments in the flexible water level scenario. The Nieuwe Keverdijkse polder north
is an exception; the water inlet increases due to the higher water levels that are maintained in the
flexible water level scenario (see Table 1). As a consequence, this polder infiltrates more to the
deep groundwater system. At the same time, seepage from the deep groundwater system
decreases to 0 mm d$^{-1}$ in the flexible water level regime (see Table 2). While the inlet flux in
Nieuwe Keverdijkse polder north increases in the flexible water level scenario, the average

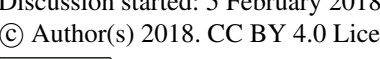



residence time still increases like in the other polders. The residence time still increases because
the total in- and outflux to and from the surface water domain are substantially reduced after
introducing the flexible water level regime.
The 2003-2011 average exchange fluxes between the surface water domain and the
groundwater domain (exfiltration and infiltration) and the field storage domain (overland flow and
inundation) for the reference and flexible water level regimes are presented in Table 3. In most
catchments, the exchange between surface water and groundwater reduces after the
introduction of flexible water level management or remains approximately the same. Exception is
again the Nieuwe Keverdijkse polder north, where infiltration increases due to the relatively high
surface water levels of the flexile water level scenario.
Overland flow and inundation are minor components in the long term average water balances of
most catchments. Due to the high groundwater and surface water levels in winter, the overland
flow contribution to surface water increases substantially in some catchments with low surface
water area percentages. Especially in Middelpolder and Ronde Hoep overland flow becomes and
important flow route in the flexible water level scenario.
To summarize the results from the solute balance modelling, Table 4 gives the 2003-2011
average surface water concentrations of Cl, N-tot, P-tot, and SO4 for the reference and the
flexible water level management scenarios. The Cl concentrations decrease in most catchments,
while N-tot concentrations generally increase. For P-tot and SO4 the results vary between the
polders. P-tot concentrations reduce in four catchments, but increase in three catchments. SO4
concentrations reduce in six catchments, but increase in three catchments.



## 4    Discussion

This research aimed at assessing the combined hydrological and hydrochemical effects of changing the water level management from a conventional fixed water level regime to a flexible, more natural regime with minimum and maximum water levels for 10 study polder catchments. Our water balance modelling results show significant changes in the water exchange fluxes between the polder catchment and the regional water system and between the groundwater, surface water, and field storage domains within the catchment. Overall, compared to the reference water level regime, the flexible water level regime scenario showed increased surface water residence times, reduced inlet and outlet fluxes, reduced groundwater-surface water exchange, and in some catchments increased overland flow. The solute balance results shows a reduction of chloride concentrations in all catchments, a reduction in P-tot and SO4 concentrations in most catchments, and an increase in N-tot concentrations in most catchments.

### 4.1    Hydrological effects

The water balance modelling results help to understand the hydrological effects of the conversion towards flexible water level management. Although the water balance models drastically simplify the real situation, they performed well in simulating the surface water and groundwater levels (see Figure 3 and Supplement B). Deviations between modelled and measured water levels can have multiple causes. For example, the actual water level management is not always just based on water levels and weather variations; in some cases human decisions such as changing the water levels for maintenance or reed harvesting play a role. Some polders (e.g. Muyeveld, Loenderveen) also receive excess water from neighboring polders, which may lead to higher water levels and fluxes than expected. In addition, we modelled all polders without taking account of the spatial differences within the polder. In most polders, the seepage or infiltration rates vary due to higher or lower water levels in neighboring polders.

The results of the water balance modelling show that the introduction of flexible water level management causes major changes in the water fluxes in all studied catchments. These





changes can be interpreted from the perspective of groundwater-surface water interactions
(Figure 4). In a conventional water level regime (Figure 4a), surface water levels in winter are
kept low, while groundwater levels are relatively high (Figure 4b). The relatively large difference
between the high groundwater levels and low surface water levels induces a relatively large
water flux from groundwater to surface water (Table 3). A relatively large outlet flux (Table 2) is
needed to compensate for the large groundwater input and maintain the low surface levels. In
summer, the high surface water levels and low groundwater levels (Figure 4c) induce a relatively
large infiltration flux from the surface water to the groundwater (Table 3). To compensate for this
infiltration flux and maintain the high surface water levels, a relatively large amount of inlet of
diverted river water is needed (Table 2). Due to the large in and out fluxes to and from the
surface water, residence times in the surface water are relatively low (Table 2). In the flexible
water level regime, the groundwater and surface water level covariate; both are relatively high in
winter (Figure 4e) and relatively low in summer (Figure 4f). The relatively small differences
between groundwater and surface water levels result in smaller exchange fluxes between
groundwater and surface water (Table 3). This also brings lower inlet fluxes in summer, lower
outlet fluxes in winter and longer residence times in surface water (Table 2).
As a result of the described changes in water fluxes, the contribution of different sources of
surface water changes as well. As a typical example, Figure 4 presents the surface water source
proportions for the Middelpolder catchment for the reference and the flexible water level regime
scenarios. The proportions are shown for a dry summer (2003), a wet summer (2007), the
average of the 2003-2011 summers, a dry winter (2004), a wet winter (2006), and the average of
the 2003-2011 winters. Figure 5 shows that the reduction in diverted river water inlet flux due to
the conversion to flexible water level management also results in a decreasing proportion of inlet
water in the catchment. The lower inlet water fractions are compensated by larger proportions of
direct precipitation water. This increase in the precipitation water proportion at the expense of the
inlet water proportion is a general consequence of the introduction of flexible water level
management in all polders.



Figure 5 also shows larger proportions of overland flow in the flexible water level management
scenario for the Middelpolder. In wet periods, surface water levels and groundwater levels are
higher in the flexible water level scenario compared to the reference scenario. In polders with
high surface water percentages, the discharged groundwater and overland flow proportions are
low in both scenarios. Inlet water and precipitation water are the dominant water sources in those
catchments.
Deep groundwater seepage comes up as a relatively small proportion after introducing the
flexible water level regime in Loenderveen Oost, Oostelijke Binnenpolder, and Westbroekse
Zodden. Up to 10% of the water volume in the Oostelijke Binnenpolder consists of deep
groundwater seepage in the dry summer of 2003. For Loenderveen Oost the deep groundwater
seepage proportion is up to 5% and for Westbroekse Zodden up to 1%. Although the proportions
are relatively low, the water quality effects of this new water source can be significant depending
on the chemistry of the seepage water (see also Yu et al., 2017).
**4.2    Water quality effects**
The changes in fluxes and the related changes in surface water source contributions help to
interpret the concentration changes due to conversion to flexible water level management
predicted by the solute balance model (Table 4). These solute balance results, however, should
be interpreted with care. The predicted concentrations are the combined contribution of all fluxes
to and from the surface water, and biochemical processes within the surface water were not
accounted for. This is not critical for the inert transport of Cl. Therefore, the Cl concentration
measurements can be used in the calibration as a check on the proportions of surface water from
different sources (Borren et al., 2012b; Hellman and Vermaat, 2012) (see for example Figure 3
for the Middelpolder). N-tot, P-tot, and SO4 concentrations, however, are also influenced by
biochemical processes such as denitrification, adsorption, sedimentation, and plant uptake (e.g.
Vermaat et al, 2016; Van der Grift et al., 2014, 2016; Yu et al., 2017). The solute balance model
also assumes that the concentrations that were assigned to the fluxes towards the surface water
do not depend on the change in water level regime. However, the predicted longer residence



times in the groundwater domain may also cause a change in the concentrations of the
groundwater exfiltration flux towards surface water. In addition, the lower summer water levels in
a flexible water level regime may enhance peat oxidation, which could also result in higher
nutrient fluxes to surface water. The difference between the predicted concentrations for the
reference and the flexible water level scenarios should therefore be regarded as an indication for
the direction of change rather than an exact quantification.
The solute balance model predicted a general decrease in Cl concentrations and a general
increase in N-tot concentrations after the conversion to flexible water level management. The
general reduction in predicted Cl concentrations is related to the reduced proportions of inlet
water. Cl concentrations in the inlet water from the regional water system are relatively high,
partly due to the discharge from several deep polders with a large brackish groundwater seepage
flux. The Cl concentrations did not change much in the polders where the Cl concentrations in
inlet water are low (Loenderveen Oost, Nieuwe Keverdijkse polder).
Although the reduced inlet also reduced the input of N-tot into the polders, the solute balance
model predicted a general increase in N-concentrations. This is related to the longer residence
times and the larger relative impact of evaporation on the water balance, which amplifies the
effects other sources. For N-tot, atmospheric input is an important remaining input for all
catchments. In some polders, higher groundwater levels in winter increased mobilization of N-tot
from the enriched upper part of the soil. The Middelpolder was the only catchment for which the
combination of changes did not cause an increase in N-tot concentrations. The decrease in N-tot
loading from inlet water and groundwater discharge in The Middelpolder was large enough to
compensate for the larger relative impact of evaporation.
For P-tot and SO4, the solute balance results varied between the polders. For most polders, the
solute balance model predicted a reduction in P-tot and SO4 concentrations after the introduction
of flexible water level management. Similar to the reduction of Cl, this is mainly related to the
reduced proportion of inlet water with relatively high P-tot and SO4 concentrations. In some



polders however, the effect of remaining P-tot or SO4 sources is amplified and counteracts the
reduced inlet. The increase in P-tot concentrations in Loenderveen Oost and the Westbroekse
Zodden is caused by increased groundwater inputs induced by the lower water levels in summer.
The increased P-tot and SO4 concentrations in the Middelpolder and the increased SO4
concentrations in the Ronde Hoep are caused by an increased mobilization from the upper part
of the soil, which is still enriched due to past agricultural activities and peat oxidation. The higher
SO4 concentrations in the Nieuwe Keverdijkse polder south are caused by the longer residence
times and the larger relative impact of evaporation. Like in the other polders, the SO4 loading
towards the surface water is substantially reduced. However, this effect counteracted in the
Nieuwe Keverdijkse polder by the large increase in residence time and associated large relative
impact of evaporation.
The effects of converting to flexible water level management on solute concentrations are
governed by the change in solute loading via water inlet, the change in residence times, and the
change in flow route contributions. Although other studies for similar catchments did not
investigate the effects of a change from fixed water levels to flexible water level management,
their results partly support the findings from this study. For example, Hellman and Vermaat
(2014) found increased nitrogen concentrations due to reduced outlet and increased evaporation
as a result of climate change. This meets with our suggestion that the longer residence times and
larger relative impact of evaporation on the water balance may cause an increase in N-tot
concentrations. Based on field-scale 2-D water transport simulations for a similar polder, Van
Beek et al. (2007) reported larger groundwater fluxes through the upper subsurface towards the
surface water as a result of an increase in surface water levels. This supports our finding that the
higher water levels in winter in a flexible water level regime may increase overland flow and
solute mobilization from the enriched upper part of the soil.





**5      Conclusions**
Changing the water level management in polders from fixed levels to more a natural, flexible
water level regime induces relevant changes in water and solute fluxes to and from the
catchment and between the groundwater, surface water, and field surface domains within the
catchment. When supported by measurements of groundwater levels, surface water levels and
fluxes, and solute concentrations, water and solute balance modelling is a feasible approach to
study the combined catchment scale effects of changing water and solute fluxes after a major
change in water level management.
Parts of the modelled effects depend on catchment specific characteristics like elevation,
seepage flux, surface water proportion, and solute concentrations. However, we found a general
increase of surface water residence times, reduction of inlet and outlet fluxes, reduction of
groundwater-surface water exchange, and in some catchments increase of overland flow. The
water quality effects of introducing flexible water level management are governed by the change
in solute loading via water inlet, the change in residence times, and the change in flow route
contributions. For Cl, the reduced input via inlet water led to lower concentrations in all polders.
However, N-tot concentrations generally increased in spite of the reduced loads via inlet water.
The effects of other N-tot sources, such as atmospheric deposition and groundwater exfiltration,
were amplified by the longer residence times and the larger relative contribution of evaporation
on the water balance. The change in P-tot and $SO_4$ concentrations varied between the
catchments as the general effect of reduced loads via inlet water was counteracted in some
catchments by increased inputs from groundwater. The modelled P-tot concentrations increased
in polders with a low surface water area percentage and P enriched top soils due to the
increased loads from overland flow towards surface water, combined with longer residence times
and increased evaporation.
This study for the first time quantified the hydrological effects and identified potential water
quality risks of changing a controlled surface water level regime. Whereas an increasing amount



of catchments are being managed in a similar way as the Dutch polders, our approach and
findings are relevant for many low-lying and often subsiding delta's around the world.
**Acknowledgements.** The authors acknowledge the entire 'Flexpeil' project team for a fruitful
cooperation. The Netherlands Enterprise Agency (RVO) is acknowledged for funding this project.



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





**Tables**
Table 1: Proportion of surface water area (relative to land surface area), land surface elevations,
approximate thickness of the Holocene peat and clay layer, reference conventional water
regimes and new flexible water level regimes for all polder catchments. Land surface elevations
and water levels in m relative to mean sea level (MSL).

| Catchment | Proportion of surface water (%) | Land surface elevation (m) | Thickness of Holocene peat and clay layer (m) | Reference conventional water level regime | Flexible water level regime |
|---|---|---|---|---|---|
| Botshol | 64 | -2.4 | 4.5-8 | Summer: -2.45, winter: -2.70 | Min: -2.65, Max: -2.45 |
| Groene Jonker | 29 | -5.0 | Ca. 4 | NA | Min: -5.60, Max: -5.10 |
| Loenderveen Oost | 98 | -1.0 | Ca. 1 | Fixed: -1.15 +/- 0.05 | Min: -1.30, Max: -1.00 |
| Middelpolder | 18 | -2.15 | Ca. 10 | Summer: -2.40, winter: -2.45 | Min: -2.55, Max: -2.25 |
| Muyeveld | 62 | -0.6 | 0-5 | Summer: -1.15, winter: -1.10 | Min: -1.20, Max: -1.05 |
| Nieuwe Keverdijkse polder north | 8 | -1.2 | 1.5-3 | Fixed: -1.65 +/- 0.05 | Min: -1.5, Max: -1.2 |
| Nieuwe Keverdijkse polder south | 8 | -1.2 | 1.5-3 | Fixed: -1.65 +/- 0.05 | Min: -1.7, Max: -1.4 |
| Oostelijke Binnenpolder | 50 | -0.9 | 0-1 | Summer: -1.25, winter: -1.35 | Min: -1.40, Max: -1.20 |
| Ronde Hoep | 4 | -2.3 | Ca. 8 | Summer: -2.85, winter: -3.00 | Min: -2.80, Max: -2.45 |
| Westbroekse Zodden | 90 | -0.7 | 0-1 | Summer: -1.00, winter: -1.05 | Min: -1.1, Max: -0.95 |

Table 2: The 2003-2011 average modelled residence times in surface water and exchange fluxes
between the surface water domain and the regional water system (inlet and outlet) and the deep
groundwater system (seepage and infiltration) for all research catchments for the reference
conventional water level regime (Ref) and the flexible water level regime (Flex). Positive fluxes
are into the surface water domain, negative fluxes leave the surface water domain.

| | Residence time (d) | | Regional Exchange | | | | Deep groundwater exchange | | | |
|---|---|---|---|---|---|---|---|---|---|---|
| | | | Inlet (mm d$^{-1}$) | | Outlet (mm d$^{-1}$) | | Seepage (mm d$^{-1}$) | | Infiltration (mm d$^{-1}$) | |
| Catchment | Ref | Flex | Ref | Flex | Ref | Flex | Ref | Flex | Ref | Flex |
| Botshol | 275 | 329 | 2.91 | 1.85 | -0.95 | 0.00 | 0.00 | 0.00 | -1.12 | -1.08 |
| Groene Jonker | NA | 275 | NA | 0.00 | NA | -0.08 | NA | 0.07 | NA | -0.04 |
| Loenderveen Oost | 806 | 874 | 0.50 | 0.11 | -0.18 | 0.00 | 0.00 | 0.14 | -0.67 | -0.61 |
| Middelpolder | 56 | 82 | 1.99 | 0.70 | -4.33 | -2.88 | 0.00 | 0.00 | -0.39 | -0.41 |
| Muyeveld | 435 | 524 | 1.54 | 0.87 | -1.13 | -0.44 | 0.00 | 0.00 | -0.75 | -0.76 |
| Nieuwe Keverdijkse polder north | 60 | 175 | 0.58 | 1.77 | -8.21 | -0.10 | 0.13 | 0.00 | -0.08 | -1.07 |
| Nieuwe Keverdijkse polder south | 29 | 78 | 0.00 | 0.00 | -11.43 | -3.99 | 0.59 | 0.19 | 0.00 | -0.35 |
| Oostelijke Binnenpolder | 126 | 209 | 2.70 | 0.00 | -1.47 | -0.04 | 0.00 | 0.27 | -0.58 | -0.46 |
| Ronde Hoep | 25 | 76 | 2.54 | 1.36 | -16.73 | -6.45 | 0.00 | 0.00 | -0.43 | -0.53 |
| Westbroekse Zodden | 365 | 511 | 1.03 | 0.00 | -1.39 | 0.00 | 0.00 | 0.03 | -0.19 | -0.31 |



Table 3: The 2003-2011 average modelled exchange fluxes between the surface water domain
and the groundwater domain (exfiltration and infiltration) and the field storage domain (overland
flow and inundation) for all research catchments for the reference conventional water level
regime (Ref) and the flexible water level regime (Flex). Positive fluxes are into the surface water
domain, negative fluxes leave the surface water domain.

| | Groundwater exchange | | | | Field storage exchange | | | |
|---|---|---|---|---|---|---|---|---|
| | Exfiltration (mm d⁻¹) | | Infiltration (mm d⁻¹) | | Overland flow (mm d⁻¹) | | Inundation (mm d⁻¹) | |
| Catchment | Ref | Flex | Ref | Flex | Ref | Flex | Ref | Flex |
| Botshol | 0.03 | 0.01 | -1.28 | -1.18 | 0.01 | 0.00 | 0.00 | 0.00 |
| Groene Jonker | NA | 0.17 | NA | -0.35 | NA | 0.00 | NA | 0.00 |
| Loenderveen Oost | 0.02 | 0.02 | -0.02 | -0.02 | 0.00 | 0.00 | 0.00 | 0.00 |
| Middelpolder | 3.37 | 1.50 | -1.23 | -0.75 | 0.16 | 1.41 | 0.00 | 0.00 |
| Muyeveld | 0.40 | 0.34 | -0.41 | -0.35 | 0.00 | 0.00 | 0.00 | 0.00 |
| Nieuwe Keverdijkse polder north | 7.54 | 0.53 | -0.38 | -1.60 | 0.00 | 0.06 | 0.00 | 0.00 |
| Nieuwe Keverdijkse polder south | 10.55 | 4.07 | -0.13 | -0.37 | 0.00 | 0.04 | 0.00 | 0.00 |
| Oostelijke Binnenpolder | 0.40 | 0.44 | -1.47 | -0.62 | 0.00 | 0.00 | 0.00 | 0.00 |
| Ronde Hoep | 15.91 | 3.58 | -1.68 | -1.66 | 0.00 | 3.48 | 0.00 | -0.19 |
| Westbroekse Zodden | 0.22 | 0.25 | -0.03 | -0.32 | 0.00 | 0.00 | 0.00 | 0.00 |


Table 4: The 2003-2011 average modelled concentrations in surface water of Chloride (Cl), Total
Phosphorus (P-tot), Total Nitrogen (N-tot), and Sulfate (SO4) for all research catchments for the
reference conventional water level regime (Ref) and the flexible water level regime (Flex).

| | Cl (mg L⁻¹) | | P-tot (mg L⁻¹) | | N-tot (mg L⁻¹) | | SO₄ (mg L⁻¹) | |
|---|---|---|---|---|---|---|---|---|
| Catchment | Ref | Flex | Ref | Flex | Ref | Flex | Ref | Flex |
| Botshol | 836 | 804 | 0.06 | 0.06 | 3.0 | 3.4 | 39 | 34 |
| Groene Jonker | NA | 43 | NA | 0.14 | NA | 12.6 | NA | 277 |
| Loenderveen Oost | 35 | 35 | 0.07 | 0.14 | 4.2 | 5.0 | 18 | 12 |
| Middelpolder | 270 | 200 | 1.05 | 1.29 | 3.8 | 3.7 | 109 | 126 |
| Muyeveld | 61 | 55 | 0.07 | 0.07 | 3.2 | 3.7 | 30 | 25 |
| Nieuwe Keverdijkse polder north | 115 | 112 | 0.13 | 0.06 | 1.3 | 2.1 | 99 | 68 |
| Nieuwe Keverdijkse polder south | 132 | 138 | 0.17 | 0.15 | 1.4 | 1.8 | 114 | 137 |
| Oostelijke Binnenpolder | 38 | 13 | 0.05 | 0.02 | 2.3 | 4.1 | 21 | 8 |
| Ronde Hoep | 231 | 186 | 0.33 | 0.30 | 4.0 | 4.3 | 127 | 149 |
| Westbroekse Zodden | 34 | 13 | 0.12 | 0.22 | 3.3 | 5.9 | 21 | 12 |

**Figures**

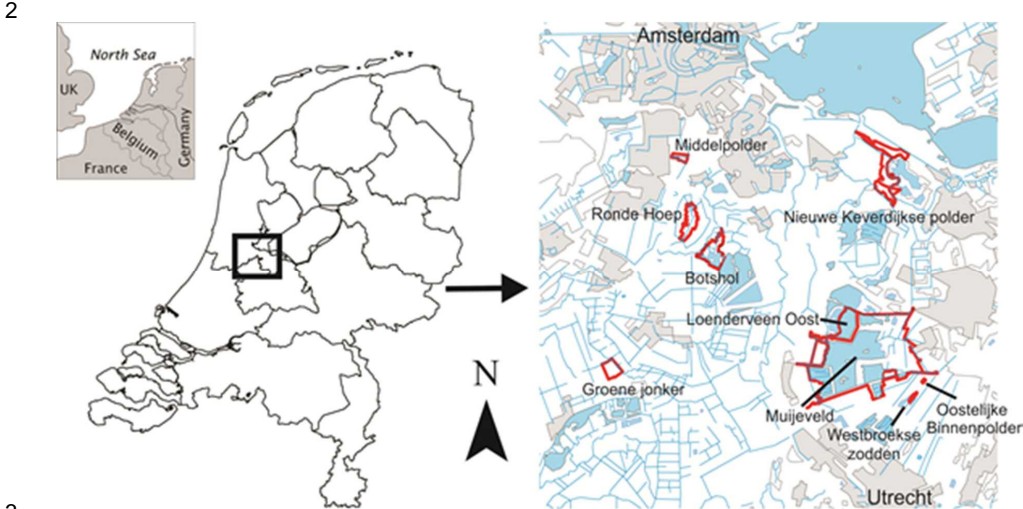

Figure 1: Study area and locations of the investigated polder catchments in The Netherlands
between Amsterdam and Utrecht.

11 Figure 2: Visualisation of the three water and solute balance domains (field storage,

12 groundwater, and surface water) and all water fluxes to and from these domains. See

13 Supplement A for the water balance model formulations.





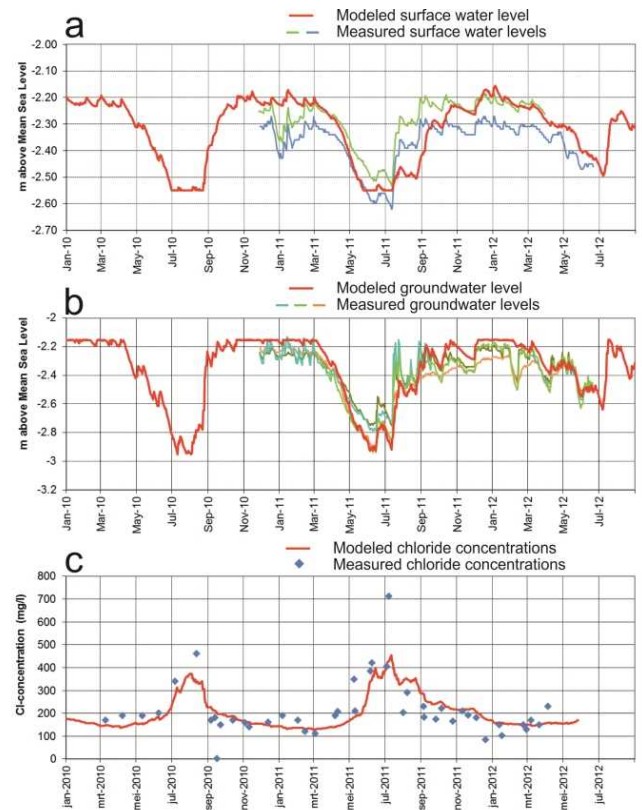

3    Figure 3: Model performance for the Middelpolder; modelled and measured surface water levels

4    (a), groundwater levels (b), and chloride concentrations (c). See Supplement B for other polders.





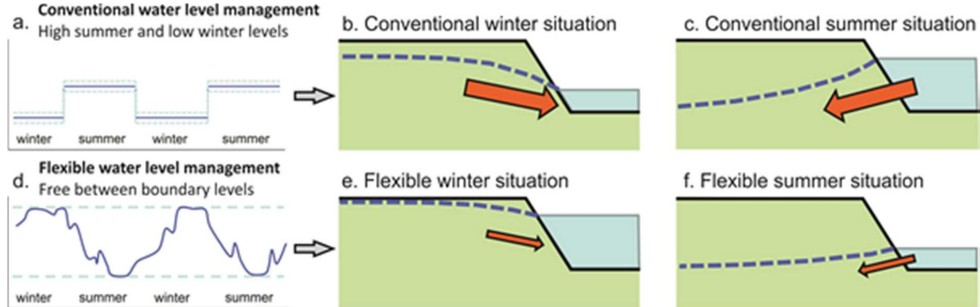

Figure 4: Groundwater-surface water interaction in conventional and flexible water level
management regimes. Fig. 4a visualises a conventional water level regime. Fig 4b and 4c show
the large fluxes from groundwater to surface water in winter (b) and from surface water to
groundwater in summer (c). Fig. 4d visualises a flexible water level regime. Fig 4e and 4f show
the relatively small fluxes from groundwater to surface water in winter (4e) and from surface
water to groundwater in summer (4f).

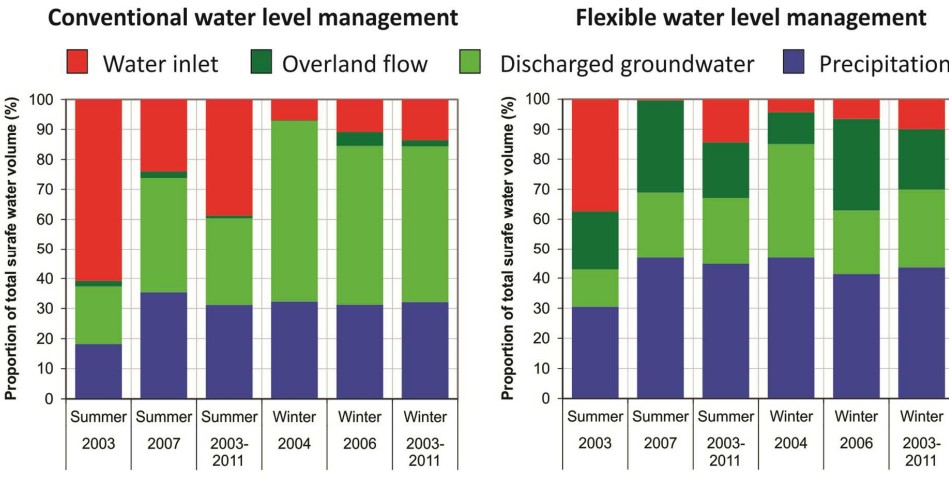

Figure 5: Proportions of water sources of surface water in the Middelpolder catchment for the
conventional water level management and the flexible water level management scenarios for a
dry summer (2003), a wet summer (2007), the average of the 2003-2011 summers, a dry winter
(2004), a wet winter (2006), and the average of the 2003-2011 winters.

