# Peer review of "Groundwater-surface water relations in regulated lowland catchments;"

_Hydrology and Earth System Sciences, 2017_

## Referee Comment (RC1) · Anonymous Referee #1 · 16 Mar 2018

General comments

This manuscript evaluates the changes in water flow, water flow paths and solute concentration when changing the water level management of several polders in the Netherlands. Water level management and connection to water quality in shallow coastal areas is a timely issue of great scientific interest and within the scope of HESS. This work presents result of quite a large experiment and combines that with water and solute balance modeling. Scientific significance and quality is good but quality of the presentation is only fair. While I acknowledge the very valuable database and efforts

behind that I am disappointed with the description of the results, discussion and conclusions. I miss a description of variability of the managed systems vs. the changes induced by the new flexible water level management. I miss a presentation of the exported solute loads. I finally miss a concluding discussion on the pro and cons of this study. At the moment it states some results but leads to nothing.

Below some more details can be found:

Abstract:

This abstract is way too long. This should be shortened to at least half the present size. What I miss is a bit more quantitative information on the water and solute flux changes induced by the new water level regimes.

Page 2 Line 11: Switching to past tense in the first sentence of the last part does not really fit.

Introduction

The introduction needs to be more concise and more consequently structured in the positive and negative sides of water level management and the lack of knowledge based on that.

P2L24: "The Netherlands" or "the Netherlands" or "Netherlands" (P4L20)? Be consequent through the manuscript.

P3L3: Is there information about the costs of the large-scale pumping vs. the costs of the damages that would occur if no pumping was done?

P3L16ff: Why do you mention the seed dispersal and germination two times in a row? Would one sentence on this not do the job?

P3L16ff: Are there distinct riparian zones in these managed catchments? I initially thought on a dominance of artificial ditches without any riparian zone around it.

[Figure]

P5L18: Every regime will have water levels between min and max. Do you mean an artificially fixed min and max? If so, clarify this in the text.

Methods

P6L17ff: Where is the hydroclimate information coming from?

P6L25: This is the first time that you mention that the study areas are reclaimed wetlands. This should be done much earlier to better focus the objectives. I thought this is also about the interference with intense agriculture...

P8L22f: I would not call nitrate and sulfate an element. ... be precise.

P8L23: Please use subscript when stating the ions.

P9L12: Was that actual ET or potential ET? If actual ET was measured how do you transfer the values to a different type of vegetation?

P9L14: But how realistic is that? Shouldn't higher variability in water levels also translate to higher variability in ET?

P9L14: You should very briefly describe how the storage of the system was assessed.

P9L25: It would make it more convincing if you state the type of averaging you did and discuss (later on) the potential errors you introduce by fixing the concentrations to the average.

Fig 3: You should spend a bit more time on this figure. Please adjust font size (labels of the panels are huge compared to the axis). On the axis it is "Cl-concentrations" on the legend "chloride concentrations" – be consistent here. What is the difference between the green and blue lines? I cannot see this stated somewhere.

Results

P11L4: This second sentences is quite useless.

P11L15f: This sentence is the same as the one in L22f and quite similar to P12L2f:

Please double read your text to avoid such things. That doesn't make the reader happy.

P12L19: When reading "solute balancing" I would rather expect to learn about the change on solute loads or concentrations and load but not concentrations alone. Overall load quantification would make sense to judge on the changes of export to downstream water resources (probably decreasing) in contrast to the concentration changes (increasing).

Discussion

P13L6: When talking about "significant" differences I would expect that you show the significance by a statistical test, which you did not do. Either show the significance or change wording to "substantial" or something similar.

P14L5f: This sentence seems to be in a wrong logical order. The outlet flux is not compensating the large groundwater influx but groundwater influx is compensating the large outlet flux. Conclusions

I expected a bit more in the conclusions than just restating the findings. So, what does that mean? Is this good or bad in terms of water quality and quantity? Does this now more resemble a natural behavior? What about the exported solute fluxes? Based on that would you recommend applying similar measures to other polders, regions? Here is a lot room for reflecting on the findings.

Fig 5: Check typos in the axis description.

---

## Referee Comment (RC2) · Anonymous Referee #2 · 11 Apr 2018

General comments The manuscript "Groundwater-surface water relations in regulated lowland catchments; hydrological and hydrochemical effects of a major change in surface water level management" aims at assessing flexible water level management schemes in contrast to a fixed water level management. In general the topic is highly interesting but I see some considerable weaknesses in the methodological approach and the amount and quality of available data. The main concern I have is the weak linkage between the monitoring efforts and the balance modelling. It seems that the two year monitoring scheme was not conducted to answer the question of this study

and hence it was not possible to use the data to clearly distinguish different operating schemes in the study catchments. It looks like that the two year period was too short. This problem is further on also relevant for the balance modelling, because a validation of the models was not carried out. It is not clear how the balance model is supported by the data. Furthermore it is not clear how the simple balance model does account for changes in residence time and the associated changes in matter transformation and fluxes, especially for an invalidated application. Therefor I cannot suggest the manuscript for publication because the given results are not fully supported by data and suggested models. A resubmission seems only be justified if the dataset can be extended and or a more process based model can be applied.

Specific comments Page 1, line 26: the type of model should be specified

Page 2, line 4: flexible water level regime water balance scenarios are unclear, please specify

Page 2, line 6 and also line 11: not clear whether this are measured or modelled results/analysis, please clarify

Page 2, line 15: questionable whether this transfer to the global scale is supported by data, because characteristics of lowland areas are highly variable at the global scale

Page 5, line 6: references are missing on this statement

Page 5: line 16-29: this paragraph should be part of the method section and not of the introduction

Page 5: line 26: it is not clear why first the intense monitoring is mentioned and one sentence later it is stated that this monitoring was not adequate to analyse changing water levels

Page 5: line 28: the objectives of the study are not clearly stated

Page 8: line 3: the monitoring was not carried out to analyse the objective of the study,

this causes problems with regard to the comparability of the findings of the different polder catchments

Page 9 line 1: what is the frequency of precipitation measurement solute measurements

Page 8: line 6-25: it is unclear where and how many groundwater, discharge and water quality measurement stations have been conducted, at least a summary table should be given

Page 9: line 27: missing surface water quality measurements is critical because it is not clear that soil solute concentrations are similar to surface water. Top soil layer can have very high nutrient concentrations although they are not only subject to transport but also to transformation on the flow path to the surface waters; this is especially true for N and P compounds.

Page 10: it is not clear how the balance model has been validated, no methodological procedures are given in the method section

Page 12, line 19: the results on the water and matter balance model are very brief and not really clear.

I did not review the discussion.

---

## Referee Comment (RC3) · Anonymous Referee #3 · 19 Apr 2018

The manuscript deals with the analysis of the effect of two water management option in low lying strongly regulated polder areas. The topic has very high concern under the view of climate change especially for coastal areas with similar conditions. They present a simple mass balance model for water balance and quality based on two years data collection extended by data from the local water boards in 10 artificial polder catchments. The study sites are located inland in the Netherlands between Amsterdam and Utrecht. The polders are characterised as low lying Marshland and were not connected. The models were calibrated based on the available data sets without

validation. The procedure is quite ambitus because these low lying polders are dominated by slightly different hydrological processes and have a higher complexity than can be expected by the first view. The two management options are the actual praxis of a fixed water level in summer and winter period and flexible management. For the last case water stage is allowed to change between specified minima and maxima levels. The manuscript is relevant for the journal. It is a quite simple analysis of the different polders which have not only for their hydrological topic importance but as well for landscape management, nature conservation, ecology in the handling of climate change induced changes in low lying areas. But it needs some improvement, reorganisation and clarifications. In the abstract they explain the model in a way that it is the major part of the manuscript, but the water balance equation are part of the appendix. The description of the solute part is completely missing. Overland flow in low land area with very small gradients has to be clarified. Based on the equations it is possibly more a kind of ponding at the surface with the source from below the surface. The equations of the solute transport and a brief description which processes are taken into account or which are missing and to which part the reliability of the model is less good are very important for the manuscript and I would suggest that the equations should be moved from the appendix to the main part. Critical discussion of the effect of greenhouse gasses should be written. Areas with low lying groundwater levels have high CO2 emission, which they have correctly mentioned. But increasing water levels can have the effect of higher methane emission with more negative effects on the environment. The introduction and method could be shortened. The ten different polders should be grouped based on the solutes of interest and possible by the dominating processes. Just based on the presented data most of the polders have Cl concentrations between 10 and 60 and then there are three with much higher ranges up to 1000 (the north western polders Botshol, Middelpolder, Ronde Hoep). Is in the mentioned three polders a different geology with faults or paleo channels / higher pressure gradients the reason as in the de Louw et al. (2010, 2013) studies? All three polders have a thick Holocene coverage. And instead the fresh water reservoir Loenderveen Oost which is

more a lake than a polder has only a layering of 1 m without high values in Cl. How can be the other solutes (nitrate, phosphorus, sulphate) categorised? Which polder systems were good represented by the mass balance equation and which less? How have they compensated the effect of the flow from polders which get additional input from the surrounding polders in case of the changing management without model validation? The models were calibrated based on their actual water management. Where there already polders under the flexible conditions? What is the error by applying a different water management to a polder under the actual conditions? How do they deal with possible changes in the hydrological processes, for example if the North-western polders with high Cl are under the influence of boils (de Louw et al.; 2010, 2013) is that dominant process under flexible conditions irrelevant? In that case the calibrated parameter set up based on fixed levels would lead to a misleading prediction of Cl by using that data set for the flexible water management conditions?

Specific comments:

P2, L15: I would be careful with the global, at a global scale these type of landscape have similar problems, but the presented model is limited to an area with comparable meteorological and geological boundaries.

P5, L16-29: shorten that paragraph and focus on a more precisely defined objective, move explaining parts to the methods

P8, L2-4: which study polder was in the two year observation period under which management option?

P9, L8: is the product information important? Would be a great benefit to add the spread sheet as supplementary material.

P9, L19-20: How was the minimum and maximum defined in case of a managed polder?

P10, L3-5: How legitimate is that procedure under the dominant geological conditions?
Please add literature.

P10, L14: Add a list of the parameters and for which process they are used and the range. Which parameters are important for solute transport? A table with quality measures (Nash, RMSE, Bias, etc.) for water balance and the different solutes would be important to judge how the different models can represent the 10 polders.

P17, L2-3: that is contradictory to the conclusion, the main source of surface water is upconing groundwater. Here is the term overland runoff misleading. Is it more ponding water?

P18, L23-25: not clear, where is the source of Phosphorus? In the polder soils or in the groundwater?

Figure 3 and the hydrographs: add a marking at which period the flexible water management started.

Appendix: some of the catchments have a very poor Cl- performance, the dynamics are fine but the level is different, why? They do not present any other chemical solute hoof the performance is for the other solutes the performance? Present quality measures for the other polders (RMS, NSE, etc.).

References:

de Louw, P.G.B., Oude Essink, G.H.P., Stuyfzand, P.J., van der Zee, S.E.A.T.M. 2010. Upward groundwater flow in boils as the dominant mechanism of salinization in deep polders, The Netherlands. Journal of Hydrology, 394, 494–506, DOI: 10.1016/j.jhydrol.2010.10.009.

de Louw, P.G.B., Vandenbohede, A., Werner, A.D., Oude Essink, G.H.P. 2013. Natural saltwater upconing by preferential groundwater discharge through boils. Journal of Hydrology, 490, 74–87, DOI: 10.1016/j.jhydrol.2013.03.025.

636, 2018.

---

## Author Comment (AC1) · 18 May 2018

General remark
We received three excellent and constructive reviews and we acknowledge all reviewers for their time and their cooperation. Based on the suggestions for improvements, we think that the manuscript has improved considerably. We wanted to state here that the reviewers recognized the complexity of the hydrologic system that was studied and brought forward quite different points for improvement. Indeed, different types of monitoring data and knowledge (soil, groundwater, surface water, water quality, hydrology) came together in the water and solute balance modeling and only a small and targeted selection of the generated output was presented. Please acknowledge our efforts to present a large-scale and integrated (groundwater-surface water, water quantity-quality) in a condense paper.

**Reviewer 1**

General comments
This manuscript evaluates the changes in water flow, water flow paths and solute concentration when changing the water level management of several polders in the Netherlands. Water level management and connection to water quality in shallow coastal areas is a timely issue of great scientific interest and within the scope of HESS. This work presents result of quite a large experiment and combines that with water and solute balance modeling. Scientific significance and quality is good but quality of the presentation is only fair. While I acknowledge the very valuable database and efforts behind that I am disappointed with the description of the results, discussion and conclusions.

We thank reviewer 1 for the thorough review and the recognition of the scientific relevance and quality. This review and the other reviews suggested excellent improvements on the presentation. We believe that the improvements mentioned in our responses below helped to improve our manuscript.

I miss a description of variability of the managed systems vs. the changes induced by the new flexible water level management.
We do not fully understand what the reviewer means with this comment. Hopefully, like the other 2 general comments below, the reviewer made this point more specific in the detailed comments and we adequately responded to this.

I miss a presentation of the exported solute loads.
We agree that most solute balancing studies focus on the loads to downstream receiving waters. Our focus, however, was on the concentration effects within the polder. The total effect on the 'receiving waters' is also complicated by the fact that some of our catchments receive more inlet water from the boezem system than they deliver discharge.  We have the loads available from the solute balance calculations, but we did not introduce this information to the paper because this was out of our scope.

To prevent similar expectations by other readers we added at the beginning of the results section:
"We only present a selection of results targeted at presenting the effects of flexible water level management on hydrology and water quality within the catchments".

I finally miss a concluding discussion on the pro and cons of this study. At the moment it states some results but leads to nothing.

Agreed, we added a paragraph to the conclusions:

" Overall, our results confirm the positive effects of flexible water level management regarding the more natural water level fluctuations, the reduced inlet water volumes needed, and the reduced discharge volumes and pumping costs. However, the expected water quality effects are more diverse, catchment specific, and uncertain. The water quality risks of within-catchment nutrient sources like nutrient-rich soils, sediments, and groundwater seepage are attenuated by flexible water level management. We therefore specifically recommend introducing flexible water level management in nature catchments where these within-catchment nutrient sources are minimal."

Below some more details can be found:

Abstract:
This abstract is way too long. This should be shortened to at least half the present size. What I miss is a bit more quantitative information on the water and solute flux changes induced by the new water level regimes.

Page 2 Line 11: Switching to past tense in the first sentence of the last part does not really fit.

We agree that the abstract is too long. We reduced the length of the abstract substantially in the revised version (from 372 to 226 words) without losing the most essential information. We also added quantitative results for the water and solute flux changes. In addition, some suggestions for improvement from the other reviewers were incorporated.

We've rewritten the abstract into:
"In lowland deltas with intensive land use, surface water levels are controlled by inlet of river water during dry periods and discharge by pumping during wet periods. The water levels are usually maintained at a fixed level year-round or at fixed winter and (higher) summer levels. Several water authorities in The Netherlands are considering a more natural and flexible regime with low levels in summer and high levels in winter. The objective of this study was to assess the catchment-scale hydrological and hydrochemical effects of such a change using water and solute balance modeling.

We focus on ten study catchments where a conversion to flexible water management was planned or recently implemented. Monitoring data from the catchments were used for validating the water balance and as boundary condition input for the solute balance calculations. For all catchments, the results show relevant changes after implementing flexible water level management. For example, the surface water residence times increased (avg. +25%), the inlet and outlet fluxes reduced (avg. -38% and -72%), the chloride concentrations reduced (avg. -14%), and the N-tot concentrations increased (avg. +13%). Both the quantification of water flux changes and the detection of water quality risks were highly relevant for the water authorities. Customizing our approach to the specific circumstances in other low-lying artificial catchments worldwide may help local water managers in optimizing their water level management."

Introduction
The introduction needs to be more concise and more consequently structured in the positive and negative sides of water level management and the lack of knowledge based on that.

We agree that the structure of the introduction is not entirely clear and consequently followed. The introduction is structured along the following steps: 1: General introduction of the topic, 2: existing knowledge on effects, 3: missing knowledge regarding effects on water quality, 4: the objective and approach of the study. In the revised introduction we moved the paragraph on the experiences with water and solute balance modelling to the objective and approach. This paragraph disconnected sections 3 and 4, which made the structure less clear. In addition, we added sentences to guide the reader through this structure. For example, after going through the positive sides we added the following sentence to introduce the text regarding potential negative sides. "Much less knowledge is available regarding the potential negative aspects of converting to a flexible water level regime."

P2L24: "The Netherlands" or "the Netherlands" or "Netherlands" (P4L20)? Be consequent through the manuscript.

Agreed and all changed to "The Netherlands".

P3L3: Is there information about the costs of the large-scale pumping vs. the costs of the damages that would occur if no pumping was done?
We agree that this would be interesting to know, but such an assessment has never been made due to the lack of knowledge about the ecological effects. A complicating factor in nature reserves is that it is hard to express the ecological damage of no pumping in terms of money. Within urban settings, a lot of flood damage to buildings and infrastructure is prevented, which is easier to quantify in money.

P3L16ff: Why do you mention the seed dispersal and germination two times in a row? Would one sentence on this not do the job?
The second and third sentences are explaining the first sentence. They explain why flexible water level management is beneficial for seed dispersal and germination. We removed the double reference to Sarneel (2014) to prevent this misunderstanding.

P3L16ff: Are there distinct riparian zones in these managed catchments? I initially thought on a dominance of artificial ditches without any riparian zone around it.
We agree that the relatively narrow and steep banks of the ditches banks in part of our research catchments are not commonly seen as 'riparian zones'. We added "and on the ditch banks" to the first sentence to avoid misunderstanding.

P5L18: Every regime will have water levels between min and max. Do you mean an artificially fixed min and max? If so, clarify this in the text.
Agreed and changed into "a flexible, more natural regime with free water level fluctuations between artificially managed minimum and maximum water levels."

Methods

P6L17ff: Where is the hydroclimate information coming from?

We added: "(based on data from the nearby Schiphol meteorological station operated by KNMI)"

P6L25: This is the first time that you mention that the study areas are reclaimed wetlands. This should be done much earlier to better focus the objectives. I thought this is also about the interference with intense agriculture. . .

Agreed. In fact, most of the western part of the Netherlands is reclaimed wetland. We added 'nature reserves' a couple of times in the revised abstract and introduction to prevent the misunderstanding that this study focuses on the effects of intense agriculture. In some of the catchments however, less intense agriculture is still in place and/or the soils are still enriched in nutrients from former intense agriculture.

P8L22f: I would not call nitrate and sulfate an element. . .. be precise.
Agreed. We changed "elements" into "solutes".

P8L23: Please use subscript when stating the ions.
Agreed and changed accordingly.

P9L12: Was that actual ET or potential ET? If actual ET was measured how do you transfer the values to a different type of vegetation?
We used reference evaporation data and used this to estimate evapotranspiration from land surfaces (mainly grassland) and open water evaporation from water surfaces using the Makking relation. We added to the text:
"The Makkink relation (Makkink, 1957) was applied to estimate grassland evapotranspiration and open water evaporation."
And there following reference was added: "Makkink, G.F., 1957. Testing the Penman formula by means of lysimeters. J. Inst. Wat. Engrs. 11, 277- 288."

P9L14: But how realistic is that? Shouldn't higher variability in water levels also translate to higher variability in ET?
We agree that changing the water level regime may also affect ET. However, the groundwater levels in these areas are always high (within 1 meter below the land surface) and the capillary rise in the clayey and peaty soils is strong. Therefore, the actual ET is usually equal or close to the potential ET. We added:
"Although changes in water levels may affect evapotranspiration, this assumption was considered legitimate for these relatively wet areas with clayey and peaty soils with strong capillary rise, where the actual evapotranspiration usually equals the potential evapotranspiration."

P9L14: You should very briefly describe how the storage of the system was assessed.
This was already described in P9 L8-11.

P9L25: It would make it more convincing if you state the type of averaging you did and discuss (later on) the potential errors you introduce by fixing the concentrations to the average.

Agreed. We used arithmetic mean concentrations. We added this in the text.
In the discussion about the uncertainties of the solute balance results we added "In addition, assigning the arithmetic mean concentrations from a limited number of measurements to the fluxes towards surface water adds to the uncertainty of the results."

Fig 3: You should spend a bit more time on this figure. Please adjust font size (labels of the panels are huge compared to the axis). On the axis it is "Cl-concentrations" on the legend "chloride concentrations" – be consistent here. What is the difference between the green and blue lines? I cannot see this stated somewhere.
Agreed, we made the suggested corrections to Figure 3.

Results
P11L4: This second sentences is quite useless.
Agreed, we deleted these sentences.

P11L15f: This sentence is the same as the one in L22f and quite similar to P12L2f: Please double read your text to avoid such things. That doesn't make the reader happy.
Agreed, we deleted the similar sentences.

P12L19: When reading "solute balancing" I would rather expect to learn about the change on solute loads or concentrations and load but not concentrations alone. Overall load quantification would make sense to judge on the changes of export to downstream water resources (probably decreasing) in contrast to the concentration changes (increasing).
We agree that most solute balancing studies focus on the loads to downstream receiving waters. Our focus, however, was on the concentration effects within the polder. The total effect on the 'receiving waters' is also complicated by the fact that some of our catchments receive more inlet water from the boezem system than they deliver discharge. We have the loads available from the solute balance calculations, but we did not introduce this information to the paper because this was out of our scope.

To prevent similar expectations by other readers we added at the beginning of the results section: "We only present a selection of results targeted at presenting the effects of flexible water level management on hydrology and water quality within the catchments".

Discussion
P13L6: When talking about "significant" differences I would expect that you show the significance by a statistical test, which you did not do. Either show the significance or change wording to "substantial" or something similar.
Agreed. We replaced significant here and at one other place in the discussion.

P14L5f: This sentence seems to be in a wrong logical order. The outlet flux is not compensating the large groundwater influx but groundwater influx is compensating the large outlet flux.

Agreed, we rephrased this sentence into: "With a larger groundwater input, a larger outlet flux (Table 2) is needed to maintain the same surface water level."

Conclusions
I expected a bit more in the conclusions than just restating the findings. So, what does that mean? Is this good or bad in terms of water quality and quantity? Does this now more resemble a natural behavior? What about the exported solute fluxes? Based on that would you recommend applying similar measures to other polders, regions? Here is a lot room for reflecting on the findings.
Agreed, we added a paragraph to the conclusions:
" Overall, our results confirm the positive effects of flexible water level management regarding the more natural water level fluctuations, the reduced inlet water volumes needed, and the reduced discharge volumes and pumping costs. However, the expected water quality effects are more diverse, catchment specific, and uncertain. The water quality risks of within-catchment nutrient sources like nutrient-rich soils, sediments, and groundwater seepage are attenuated by flexible water level management. We therefore specifically recommend introducing flexible water level management in nature catchments where these within-catchment nutrient sources are minimal."

Fig 5: Check typos in the axis description.
We corrected the typos in Figure 5.

**Reviewer 2**

General comments

The manuscript "Groundwater-surface water relations in regulated lowland catchments; hydrological and hydrochemical effects of a major change in surface water level management" aims at assessing flexible water level management schemes in contrast to a fixed water level management. In general the topic is highly interesting but I see some considerable weaknesses in the methodological approach and the amount and quality of available data.

We thank reviewer 2 for his review and recommendations. We think that our manuscript has greatly improved based on all comments.

The main concern I have is the weak linkage between the monitoring efforts and the balance modelling. It seems that the two year monitoring scheme was not conducted to answer the question of this study and hence it was not possible to use the data to clearly distinguish different operating schemes in the study catchments. It looks like that the two year period was too short. This problem is further on also relevant for the balance modelling, because a validation of the models was not carried out. It is not clear how the balance model is supported by the data.

Our findings and conclusions are based on the water and solute balance model, for which the monitoring provides input and validation data. The reference situation, the timing of the actual change in water level management, and the practical water level management varied among the study catchments. Therefore, the monitoring did not provide us with adequate data for comparing the situation before and after the implementation of flexible water level management. To enable identical catchment-scale assessments for all ten catchments, the monitoring data and system knowledge were combined in water and solute balance models for each catchment. This was not clear everywhere in the manuscript. We improved this in the revised manuscript, based on the specific comments below and the suggestions from the other reviewers.

Furthermore it is not clear how the simple balance model does account for changes in residence time and the associated changes in matter transformation and fluxes, especially for an invalidated application.

The application was validated, although we'll improve the text on the validation methodology, we'll add a table with performance indicators, and we'll add a discussion of the performance (also based on a suggestion by reviewer 3). The changes in residence times within the surface water (or groundwater) storage of the water balance model are a direct result from the calculations. Biochemical processes were not incorporated in the solute balance model. This was discussed in P15 L17-P18 L6. This part of the discussion also addresses the potential effect of the change in residence times on chemical processes and the associated uncertainty of the solute balance results.

Therefor I cannot suggest the manuscript for publication because the given results are not fully supported by data and suggested models. A resubmission seems only be justified if the dataset can be extended and or a more process based model can be applied.

Specific comments

Page 1, line 26: the type of model should be specified

The type of model was a "water and solute balance model" as was indicated in the abstract and specified in the methods section.

Page 2, line 4: flexible water level regime water balance scenarios are unclear, please specify
Agreed and changed in the revised abstract (see also above at the response to reviewer 1).

Page 2, line 6 and also line 11: not clear whether this are measured or modelled results/analysis, please clarify
Agreed. This are modelled results and we made this clear in the revised abstract (see also above at the response to reviewer 1).

Page 2, line 15: questionable whether this transfer to the global scale is supported by data, because characteristics of lowland areas are highly variable at the global scale
All agreed and changed accordingly in our rewritten and shortened abstract (see also above at the response to reviewer 1). The final sentence is a more modest statement regarding the application of our approach in other areas taking account of local conditions: "Customizing our approach to the specific circumstances in other low-lying artificial catchments worldwide may help local water managers in optimizing their water level management."

Page 5, line 6: references are missing on this statement
Agreed. We added references to the following papers:
Ellis, J. B., Marsalek, J., and Chocat, B.: Encyclopedia of Hydrological Sciences, Urban Water Quality, 1st edition, M. G. Anderson, John Wiley & Sons, Ltd, United States, 8, 97, 2005.
Yan, R., Huang, J., Li, L., Gao, J., 2017. Hydrology and phosphorus transport simulation in a lowland polder by a coupled modeling system, Environ. Pollut., 227, 613–625.

Page 5: line 16-29: this paragraph should be part of the method section and not of the introduction
In this paragraph, we introduce the objective and give a short outline of the approach. In our opinion, this information is at the correct location at the end of the introduction. The paragraph has been rewritten in the revised paper in response to a comment from reviewer 1.

Page 5: line 26: it is not clear why first the intense monitoring is mentioned and one sentence later it is stated that this monitoring was not adequate to analyse changing water levels
Agreed. We rewrote this paragraph also in response to reviewer 1. As part of the revised paragraph, we state:
"Monitoring data was collected for validating the water balance model and as boundary condition input for the solute balance calculations. The reference situation, the timing of the actual change in water level management, and the practical water level management varied among the study catchments. Therefore, the monitoring did not provide us with adequate data for comparing the situation before and after the implementation of flexible water level management. To enable identical catchment-scale assessments for all ten catchments, the monitoring data and system knowledge were combined in water and solute balance models for each catchment."

Page 5: line 28: the objectives of the study are not clearly stated
The objective was stated on page 5, line 16-18 of the original manuscript.

Page 8: line 3: the monitoring was not carried out to analyse the objective of the study, this causes problems with regard to the comparability of the findings of the different polder catchments
Agreed. See also our response to your comment on page 5, line 16. Our findings and conclusions are based on the water and solute balance model, for which the monitoring provides input and validation data. For the reasons mentioned the revised paper and in the reply to your comment on page 5, line 26, we could not rely on monitoring data alone to meet our objective.

Page 9 line 1: what is the frequency of precipitation measurement solute measurements
This was biweekly sampling. We added this information in this sentence.

Page 8: line 6-25: it is unclear where and how many groundwater, discharge and water quality measurement stations have been conducted, at least a summary table should be given
We agree that we did not give the details about the monitoring. In section 2 we described the general setup of the monitoring without going into the details for all different types of monitoring (discharges, water levels, groundwater levels, surface water quality, groundwater quality, soil moisture concentrations) within each of the 10 research catchments. This would put too much emphasis on the monitoring in this paper. Therefore, we referred to Borren et al. (2012a) and Smolders et al. (2012) for more detailed information on the monitoring in each catchment.

Page 9: line 27: missing surface water quality measurements is critical because it is not clear that soil solute concentrations are similar to surface water. Top soil layer can have very high nutrient concentrations although they are not only subject to transport but also to transformation on the flow path to the surface waters; this is especially true for N and P compounds.
We agree that monitoring information of solute concentrations in overland flow, and also discharge volumes of overland flow would have been valuable. Note that we did collect surface water quality measurements at many locations as was described in section 2.2. Overland flow was not recognized as an important flow route in the relatively flat lowland research catchments. Our water balance results also suggest that overland flow did not occur in most catchments (see Table 3). However, in two catchments (Ronde Hoep and Middelpolder) overland flow appeared to become substantial after introducing flexible water level management. Only the results for these 2 catchments are affected by our assumption of concentrations in overland flow being equal to the concentrations in soil moisture at 25 cm below the surface. We think that this assumption is a legitimate one. The soils in the 2 catchments are still in agricultural use, and the upper 30 cm of the soil is mixed every few years by tillage. This means that ponds and water in the upper 5 cm is subjected to the same nutrient rich conditions as water and soil moisture at a depth of 25 cm.
To justify our assumption we added to the text: "This soil moisture interacts with the same nutrient-rich soil as overland flow, because the upper 30 cm of the soil was mixed by tillage during former intensive agricultural land use."

Page 10: it is not clear how the balance model has been validated, no methodological procedures are given in the method section
Agreed. The validation was done by comparison of groundwater water level and surface water levels. In the revised paper, we will add a table with the water balance performance indicators and add a discussion paragraph on these performances (also suggested by reviewer 3).

Page 12, line 19: the results on the water and matter balance model are very brief and not really clear.

The results section just presents the results (facts) without interpretation. The interpretation (what do the results tell us about the objective) is described in the discussion section. We think that this is the best and most common structure of a scientific paper.

I did not review the discussion.

We regret this, because the discussion is where the interpretation of the results is presented, which this reviewer expected in the results section.

Reviewer 3

The manuscript deals with the analysis of the effect of two water management option in low lying strongly regulated polder areas. The topic has very high concern under the view of climate change especially for coastal areas with similar conditions. They present a simple mass balance model for water balance and quality based on two years data collection extended by data from the local water boards in 10 artificial polder catchments. The study sites are located inland in the Netherlands between Amsterdam and Utrecht. The polders are characterised as low lying Marshland and were not connected. The models were calibrated based on the available data sets without validation. The procedure is quite ambitus because these low lying polders are dominated by slightly different hydrological processes and have a higher complexity than can be expected by the first view. The two management options are the actual praxis of a fixed water level in summer and winter period and flexible management. For the last case water stage is allowed to change between specified minima and maxima levels. The manuscript is relevant for the journal. It is a quite simple analysis of the different polders which have not only for their hydrological topic importance but as well for landscape management, nature conservation, ecology in the handling of climate change induced changes in low lying areas.

We thank reviewer 3 for the thorough review and the recognition of the scientific relevance and the complexity of the hydrology of polder systems. Several revisions based on this review really improved the quality of our manuscript.

But it needs some improvement, reorganisation and clarifications.

In the abstract they explain the model in a way that it is the major part of the manuscript, but the water balance equation are part of the appendix. The description of the solute part is completely missing.
Agreed. We've rewritten the abstract, also based on suggestions by reviewer 1. The description of both the water and solute balance is in section 2.3. To keep the main paper concise, we present the parameter list and the model formulations not in the main text but in Supplement A. This enabled us to be both complete in the presentation of the water and solute balance model, while keeping the main text clear for readers who have less interest in these details.

Overland flow in low land area with very small gradients has to be clarified. Based on the equations it is possibly more a kind of ponding at the surface with the source from below the surface. The equations of the solute transport and a brief description which processes are taken into account or which are missing and to which part the reliability of the model is less good are very important for the manuscript and I would suggest that the equations should be moved from the appendix to the main part.
We agree that this type of overland flow needs some clarification in the main text. In these flat landscapes, ponds form on the land surface in wet periods. At a certain moment, these ponds start to flow over directly into the surface water. This ponding and overland flow can both occur due to infiltration excess and saturation excess, which is both accounted for in our model.
We added to the text: "Overland flow in these flat areas usually starts with ponds on the field that, after more precipitation, start to flow over directly into the surface water. Like overland flow in

higher gradient landscapes, this type of overland flow can be caused by infiltration excess or saturation excess, which is both accounted for in the water and solute balance model."
See our answer to the previous comment for our reasoning behind placing the model formulations in a Supplement.

Critical discussion of the effect of greenhouse gasses should be written. Areas with low lying groundwater levels have high CO2 emission, which they have correctly mentioned. But increasing water levels can have the effect of higher methane emission with more negative effects on the environment.

Increased NH4 emissions are indeed a risk of higher water levels, and increased CO2 emissions are indeed a risk of lower groundwater levels. This has been the topic of another work package within the project Hendriks et al. (2012). We added the main findings of this study to the introduction: "Another example is the potential increase in greenhouse gas emissions. The lower water levels in summer may cause enhanced peat oxidation and CO2 emissions, while the wet conditions in winter may cause emission of CH4. However, concluded that greenhouse gas emissions are mainly governed by the average surface water levels, rather than the seasonal or daily variations in water levels."

The introduction and method could be shortened.

We agree that a shorter introduction and methods section could be an improvement. However, we also think that these sections already have a high information density, which reflects the complexity of the hydrology that this reviewer recognized. We have shortened parts of the abstract, introduction, and methods section without losing essential information. Still, extra explanations requested by the reviewers have not made the overall paper shorter.

The ten different polders should be grouped based on the solutes of interest and possible by the dominating processes. Just based on the presented data most of the polders have Cl concentrations between 10 and 60 and then there are three with much higher ranges up to 1000 (the north western polders Botshol, Middelpolder, Ronde Hoep). Is in the mentioned three polders a different geology with faults or paleo channels / higher pressure gradients the reason as in the de Louw et al. (2010, 2013) studies? All three polders have a thick Holocene coverage. And instead the fresh water reservoir Loenderveen Oost which is more a lake than a polder has only a layering of 1 m without high values in Cl. How can be the other solutes (nitrate, phosphorus, sulphate) categorised? Which polder systems were good represented by the mass balance equation and which less?

There are a lot of different catchment characteristics on which a division in groups could be based. The recognition of the most essential characteristics is rather a result of this study then available knowledge beforehand. Therefore, we have not grouped the catchments in the methods and results section; in the tables they are listed in alphabetic order. In the discussion, the catchments are often mentioned in groups, because a specific effect was similar in those catchments. However, also in the discussion the grouping depends on the specific effect that was described. For these reasons, we're not confident about one overall grouping of catchment types.

In the revised paper, we will add a table with the water balance performance indicators and add a discussion paragraph on these performances.

How have they compensated the effect of the flow from polders which get additional input from the surrounding polders in case of the changing management without model validation? The models were calibrated based on their actual water management. Where there already polders under the flexible conditions? What is the error by applying a different water management to a polder under the actual conditions?

The water and solute balance models were all calibrated towards the water level management that was in place during the monitoring period. To make this clear in the text, we added:
"The actual water level regimes during the monitoring period were used during this calibration."

How do they deal with possible changes in the hydrological processes, for example if the North-western polders with high Cl are under the influence of boils (de Louw et al.; 2010, 2013) is that dominant process under flexible conditions irrelevant? In that case the calibrated parameter set up based on fixed levels would lead to a misleading prediction of Cl by using that data set for the flexible water management conditions?

In our water balance model approach we neglect spatial differences within our research catchments. As indicated by De Louw, the groundwater inputs can be highly variable in space (boils). Therefore, distributed modelling would be needed to study spatial patterns within the catchment. The spatial variations do not necessarily have impact on the changes in the total fluxes to and from the catchments and between groundwater and surface water. We chose the non-distributed water and solute balance approach to be able to integrate all possible flow routes and interactions between land and water and between hydrology and hydrochemistry without using too complicated models with long calculation times. As shown before by Hellman and Vermaat (2012) and Kieckbusch and Schrautzer (2007), water and solute balance models are effective for disentangling the combined effect of (changes in) multiple sources and pathways of water and solutes in similar catchments.

Specific comments
P2, L15: I would be careful with the global, at a global scale these type of landscape have similar problems, but the presented model is limited to an area with comparable meteorological and geological boundaries.

Agreed and changed accordingly in our rewritten and shortened abstract (see also above at the response to reviewer 1). The final sentence is a more modest statement regarding the application of our approach in other areas taking account of local conditions: "Customizing our approach to the specific circumstances in other low-lying artificial catchments worldwide may help local water managers in optimizing their water level management."

P5, L16-29: shorten that paragraph and focus on a more precisely defined objective, move explaining parts to the methods

Agreed. In this paragraph, we introduced the objective (first sentence) and give a short outline of the approach. The paragraph has been rewritten in the revised paper also in response to a comment from reviewer 1.

P8, L2-4: which study polder was in the two year observation period under which management option?

Agreed. To provide this information we added the moment of introduction of flexible water level management in each catchment to Table 1 and we referred to this information in the text.

P9, L8: is the product information important? Would be a great benefit to add the spread sheet as supplementary material.

Great idea to add the Excel spreadsheet as supplementary material. We will do that and we will add a reference to this supplement. We're quite used to mentioning the software used in papers.

P9, L19-20: How was the minimum and maximum defined in case of a managed polder?

These were the actual minimum and maximum water levels that are maintained by the water board by water inlet and pumping. These levels were presented in Table 1. These are not defined from data, the minimum and maximum water levels are decisions of the water authorities together with stakeholders based on land use and hydrology. To avoid misunderstanding, we added to the text: " See Table 1 for the water levels in each catchment before and after the introduction of flexible water level management, which were also applied in model scenario's."

P10, L3-5: How legitimate is that procedure under the dominant geological conditions? Please add literature.

The deep piezometers (3-8 in each catchment depending on their size) were placed in the fluvial sand layer, just below the cover layer of peat and sand. The spatial and temporal variability in nutrient concentrations is limited in this aquifer, as it is protected from human and climatic variations by the cover layer. The concentrations within the aquifer below the cover layer have been used to characterise seepage water concentrations in other studies as well. A recent example is the paper by Yu et al., 2017. To clarify this, we added the following sentences:
"These deep piezometers were installed in the sandy aquifers below the cover layer of sand and peat. Protected from human and climatic variations by the cover layer, the spatial and temporal variations in this aquifer are limited. Therefore, assigning average concentrations from 3-8 deep piezometers to seepage water was considered legitimate and was applied before by e.g. Yu et al., 2017."

P10, L14: Add a list of the parameters and for which process they are used and the range. Which parameters are important for solute transport? A table with quality measures (Nash, RMSE, Bias, etc.) for water balance and the different solutes would be important to judge how the different models can represent the 10 polders.

Great idea to add a table with model quantitative model performance measures. We will add this table to the revised paper.
The list of parameters was presented in Appendix A. Given the length of this list, we've chosen to add this to this Appendix A, which also contains all model formulations of the water and solute balance model. The most important and sensitive parameters for the water and solute balance modelling were presented in the text (P10, L13-16 of the original paper)

P17, L2-3: that is contradictory to the conclusion, the main source of surface water is upconing groundwater. Here is the term overland runoff misleading. Is it more ponding water?

In the conclusions we stated (P18 I20-22 of the original paper): "The change in P-tot and SO¬4 concentrations varied between the catchments as the general effect of reduced loads via inlet water was counteracted in some catchments by increased inputs from groundwater."
This conclusion was partly based on the observation in P17, L2-3 where we stated: "The increase in P-tot concentrations in Loenderveen Oost and the Westbroekse Zodden is caused by increased groundwater inputs induced by the lower water levels in summer."
We think that both statements in the discussion and in the conclusions are in line with each other.

We added explanation to the revised paper about the process of overland flow in flat landscapes (see our response to one of the general remarks).

P18, L23-25: not clear, where is the source of Phosphorus? In the polder soils or in the groundwater?
There is a legacy store of P in the soils. Surface runoff and upper groundwater flow take up P from this store and transport it to the surface water. We think that we wanted to combine too much information within one sentence. We rephrased the rather complicated sentence into:
"The modelled P-tot concentrations increased in polders with a low surface water area percentage and P enriched top soils. In these cases, the increased overland flow and shallow groundwater flux transported more P to surface water. The concentration effect of this internal source was again amplified by the longer residence times and increased evaporation."

Figure 3 and the hydrographs: add a marking at which period the flexible water management started.
For most catchments, the moment of conversion to flexible water management was outside the monitoring period. In the revised paper, we've added the moments of the introduction of flexible water level management to Table 1.

Appendix:
some of the catchments have a very poor Cl- performance, the dynamics are fine but the level is different, why? They do not present any other chemical solute hoof the performance is for the other solutes the performance? Present quality measures for the other polders (RMS, NSE, etc.).
The good aspect of using Cl for checking the water balance is that Cl is conservative. In addition, in some of the catchments, the concentrations of inlet water were very high, which makes Cl a good tracer to make a distinction between inlet water and precipitation water impacts. In some other catchments, the differences between the water main sources are less clear. Especially in these catchments, the uncertainty in the assigned Cl concentrations and the possible influence of unknown sources of Cl reduces the Cl-performance.
In the revised paper, we will add a table with the water balance performance indicators and add a discussion paragraph on these performances.

References:
de Louw, P.G.B., Oude Essink, G.H.P., Stuyfzand, P.J., van der Zee, S.E.A.T.M. 2010. Upward groundwater flow in boils as the dominant mechanism of salinization in deep polders, The Netherlands. Journal of Hydrology, 394, 494–506, DOI: 10.1016/j.jhydrol.2010.10.009.

de Louw, P.G.B., Vandenbohede, A., Werner, A.D., Oude Essink, G.H.P. 2013. Natural saltwater upconing by preferential groundwater discharge through boils. Journal of Hydrology, 490, 74–87, DOI: 10.1016/j.jhydrol.2013.03.025.